# The Relevance of Implanted Percutaneous Electrical Nerve Stimulation in Orthopedics Surgery: A Systematic Review

**DOI:** 10.3390/jcm13133699

**Published:** 2024-06-25

**Authors:** Sarthak Parikh, Alexandra C. Echevarria, Brandon R. Cemenski, Travis Small

**Affiliations:** 1Saint Francis Health Systems, Tulsa, OK 74136, USA; mcheruvu@saintfrancis.com; 2Department of Orthopedic Surgery, Oklahoma State University, Tulsa, OK 74078, USA; 3Orthopedic and Trauma Services of Oklahoma, Tulsa, OK 74135, USA; 4Kiran Patel College of Osteopathic Medicine, Nova Southeastern University, Davie, FL 33328, USA; ae934@mynsu.nova.edu; 5College of Osteopathic Medicine, Des Moines University, Des Moines, IA 50266, USA; brceme15@student.dmu.edu

**Keywords:** peripheral nerve stimulation, pain management, arthroplasty, orthopedics

## Abstract

**Background**: Percutaneous peripheral nerve stimulation (PNS) is a form of neuromodulation that involves the transmission of electrical energy via metal contacts known as leads or electrodes. PNS has gained popularity in orthopedic surgery as several studies have supported its use as a pain control device for patients suffering from pain due to orthopedic pathologies involving the knee, shoulder, and foot. The purpose of this systematic review is to summarize the literature involving peripheral nerve stimulation in orthopedic surgery. The existing body of literature provides support for further research regarding the use of PNS in the management of knee pain, hip pain, shoulder pain, foot pain, and orthopedic trauma. Notably, the evidence for its efficacy in addressing knee and shoulder pain is present. **Methods**: This study was conducted following PRISMA guidelines. Seven hundred and forty-five unique entries were identified. Two blinded reviewers assessed each article by title and abstract to determine its relevance and categorized them as “include”, “exclude”, and “maybe”. After a preliminary review was completed, reviewers were unblinded and a third reviewer retrieved articles labeled as “maybe” and those with conflicting labels to determine their relevance. Twenty-eight articles were included, and seven hundred and seventeen articles were excluded. Articles discussing the use of PNS in the field of orthopedic surgery in patients > 18 years of age after 2010 were included. Exclusion criteria included neuropathic pain, phantom limb pain, amputation, non-musculoskeletal related pathology, non-orthopedic surgery related pathology, spinal cord stimulator, no reported outcomes, review articles, abstracts only, non-human subjects. **Results**: A total of 16 studies analyzing 69 patients were included. All studies were either case series or case reports. Most articles involved the application of PNS in the knee (8) and shoulder (6) joint. Few articles discussed its application in the foot and orthopedic trauma. All studies demonstrated that PNS was effective in reducing pain. **Discussion**: Peripheral nerve stimulation can be effective in managing postoperative or chronic pain in patients suffering from orthopedic pathology. This systematic review is limited by the scarcity of robust studies with substantial sample sizes and extended follow up periods in the existing literature.

## 1. Introduction

Electrical percutaneous peripheral nerve stimulation (PNS) is a form of neuromodulation which involves the transmission of electrical energy via metal contacts known as leads or electrodes [1]. Several mechanisms attempt to explain the effectiveness of PNS. The pathogenetic basis of electrical nerve stimulation was substantiated using the theory of gate control of pain, a theory proposed by Melzak and Wall in 1965, which explains that the sensory input by large Aβ fibers obstructs the transmission of input from the small pain fibers. This theory suggests that stimulation of large nonnociceptive afferent fibers (pressure, touch, vibration) will “close” the dorsal horn interneurons and inhibit pain perception, whereas small nociceptive afferent fibers open the dorsal horn interneurons [2,3,4,5,6,7,8]. In other words, the low activation threshold for the large Aβ fibers causes early activation and resultant inhibition of nociceptive Aδ and C pain fibers by exciting the associated dorsal horn interneuron involved in processing and transmitting pain [5,9]. PNS works by providing a nonpainful stimulation to a peripheral nerve resulting in an increased electrical threshold, decreased excitability, and the slowing of conduction velocity, which inhibits pain [2,3,4,5,6,7,8]. Electrical stimulation of these fibers can downregulate neurotransmitters (substance P and calcitonin-gene-related peptide [CGRP]), endorphins, and local inflammatory mediators associated with pain and a nerve’s sensory distribution [5,9]. It is the release of these neurotransmitters that augments the pain response leading to vasodilation, plasma extravasation, macrophage chemotaxis, and/or mast cell degranulation [6,10]. In 2020, a study by Lin et al. confirmed that the nonpainful electrical stimulation of the nerves through needle electrodes can successfully decrease pain perception to the face [11]. The clinical and experimental use of electrical stimulation for pain relief is a field of medicine that continues to grow.

In modern applications of PNS, an external generator unit is connected to a lead placed 0.5 to 3.0 cm from the targeted nerve. The generator controls the frequency, amplitude, and pulse duration of the electrical stimulation to provide individualized treatment [9,12,13]. These are programmed by the person providing the device. The application of PNS typically involves using a continuous biphasic waveform with a pulse duration ranging from 250 to 500 ms, and it can be delivered at either low or high frequency. Temporary percutaneous PNS trials are often implemented prior to permanent PNS leads and last up to 60 days [12].

Although PNS is an acceptable nonsurgical form of pain management, hardware and biological complications can occur. Leads can break or become dislodged, while electrical stimulation can cause muscle spasms and discomfort. Removal of the leads often rectifies these complications. Risk of infection, although low, is also present as these devices are placed subcutaneously and require a skin incision [13].

PNS boasts a diverse range of applications, addressing nerves implicated in conditions such as paralysis, epilepsy, depression, urinary and gastrointestinal disorders, headaches, and chronic pain syndromes related to amputation, post stroke shoulder pain, complex regional pain syndrome and neuropathies [1,14]. Recently, PNS has gained popularity in orthopedic surgery as several feasibility studies and proof-of-concept case reports and case series have supported its use as a pain control device in patients suffering from pain due to orthopedic pathologies involving the knee, shoulder, and foot [15,16,17]. It has been shown to reduce pain following surgery but can also be used to treat pain in patients when surgery is contraindicated.

The purpose of this systematic review is to summarize the literature involving PNS in orthopedic surgery and provide a comprehensive conclusion about its role. It is hypothesized that PNS may be a feasible treatment modality for the management of pain in patients suffering from pathologies managed by orthopedic surgeons. The use of percutaneous PNS by orthopedic surgeons for orthopedic pathologies is a relatively new application of neuromodulation. If the current literature demonstrates that PNS can deliver adequate pain relief for patients suffering from orthopedic conditions, then orthopedic surgeons can consider employing this treatment modality to optimize their patients’ pain control and improve their outcomes.

## 2. Materials and Methods

Institutional review board (IRB) approval determined this study does not fall under human subject research and does not need further oversight. The systematic review was conducted using Rayyan.ai and the Preferred Reporting Items for Systematic Reviews and Meta-Analyses (PRISMA) guidelines [18,19]. Rayyan is an online platform designed for researchers to conduct systematic literature reviews and other types of evidence synthesis. It provides a centralized hub for managing the entire review process [19]. The purpose of this review is to provide a concise summary of studies that explore the role of PNS in the field of orthopedic surgery. The data was collected by first using the following search strings to identify articles on Cochrane, Ovid, PubMed, Embase and Scopus: “peripheral nerve stimulation” AND “pain”, “peripheral nerve stimulation” AND “osteoarthritis”, “peripheral nerve stimulation” AND “orthopedics”. Seven hundred and forty-five unique entries were identified. The search was conducted on 23 November 2023. Two blinded reviewers assessed each article by title and abstract to determine their relevance and categorized them as “include”, “exclude” and “maybe”. The PICO (population, intervention, comparator, and outcome) framework was used to include and exclude articles [20]. Articles reporting outcomes discussing the use of PNS in the field of orthopedic surgery in patients >18 years of age after 2010 were included. Exclusion criteria include neuropathic pain, phantom limb pain, amputation, non-orthopedic surgery related pathology, spinal cord stimulator, no reported outcomes, review articles, abstracts only, non-human subjects (Figure 1). Outcomes regarding pain, function, and opioid consumption were tracked. After a preliminary review was completed, reviewers were unblinded and a third reviewer retrieved articles labeled as “maybe” and those with conflicting labels to determine their relevance. Sixteen articles were included, and seven hundred seventeen articles were excluded. The final list of 16 articles were then categorized into the following fields of orthopedic surgery: hip and knee, shoulder, and foot. All included articles were either case reports or case series. The methodological quality and synthesis of case series and case studies were assessed using the questionnaire published by Murad et al. [21].

Data synthesis and tabulation was carried out using Microsoft Excel 2024. Each article was analyzed by type of study, procedure, population, indication, nerve targeted, and outcomes. Results were then compared, contrasted, and compiled into tables for evaluation. Articles were categorized and summarized in subsections based on location (hip and knee, shoulder, and foot). Only one study included statistics involving mean differences. A meta-analysis could not be carried out due to the variability in reported outcomes and follow up times. Outcome reporting varied and reports were not standardized between studies, making statistical analysis between studies impractical. The protocol was not prepared.

PNS is a Federal Drug Administration (FDA) approved implant that is placed percutaneously adjacent to the nerve of interest. All case series included were IRB approved. Patient consent was obtained in all studies. Multiple companies have produced the implantable device. The system is comprised of a percutaneous lead that has an open-coil design. It is preloaded in a 20-gauge introducer needle and inserted percutaneously using ultrasound guidance. Leads are then connected to an external pulse generator programmed to deliver stimulation at a specific frequency and amplitude that can be personalized. PNS trials are performed for 60 days prior to offering the patient a permanent lead placement [16].

PNS can also be delivered by pressing electrodes onto and anterior to the ear, targeting branches from cranial nerves V, VII, IX, and X and the occipital and great auricular nerves. A wearable pulse generator is also placed adjacent to the ear to control stimulation delivery. This device is also cleared by the FDA to reduce symptoms of opioid withdrawal. The device is small, disposable, and easily adherent to the skin [22].

## 3. Results

### 3.1. Hip and Knee

Eight case series examined the application of PNS in the context of knee pain [16,22,23,24,25,26,27,28]. One study focused on patients with a patellar tendon autograft anterior cruciate ligament (ACL) reconstruction and seven studies investigated PNS for the management of pain due to osteoarthritis of the knee or postoperative pain after total knee arthroplasty (TKA) or total hip arthroplasty (THA) [11]. Collectively, 35 patients were analyzed. All studies reported pain-related outcomes. Leads in these investigations targeted the femoral nerve, sciatic nerve, auricular nerve, and/or saphenous nerve via ultrasound. The purpose of these studies was to demonstrate the efficacy of percutaneous PNS as a treatment modality for patients with contraindications to surgery or for the management of postoperative pain. Seven studies documented the generator setting frequency, amplitude, and pulse width as 100 Hz, between 0.2–20 mA and between 15–200 µs, respectively. Exact measurements were modified to the patient’s desire. Patients rated their pain on a numerical scale from 0 to 10 (0 being no pain and 10 being the worst pain).

In two studies, PNS was used to address osteoarthritic knee pain in four patients with contraindications to surgery due to young age and/or an elevated BMI (Table 1a) [27,28]. Patients rated their knee pain on a scale from 0 to 10 before and after the intervention; however, not all post intervention responses were documented. Three patients experienced immediate pain relief upon lead placement and stimulation. These patients reported enhancements in their functional capacities such as activities of daily living and job performance [10,16]. Conversely, a single patient experienced muscle contractions with femoral nerve stimulation and therefore was only treated with saphenous nerve stimulation [10]. However, two weeks after the procedure this patient reported uncomfortable cramping and inadequate coverage of the pain area and the lead was eventually removed [10]. The lack of discussion regarding the selection process by Zhu et al. could potentially introduce selection bias [28]. Additionally, the patient described by Hasoon et al. opted for PNS instead of surgery, which might contribute to response bias [27].

Four case series discussed the use of PNS in 21 patients with knee pain following TKA (Table 1b). PNS was placed either before surgery or up to 97 days after [16,23,24,25]. In 7 patients with leads placed before surgery, stimulation was delivered continuously until the day of surgery, disconnected before surgery, reconnected 20 h after surgery and continuously turned on for up to 6 weeks [16]. A single-injection adductor canal block was also administered. During the first week six patients rated their overall pain as mild (<4/10), which was sustained until week 4 [16]. The median time to opioid cessation was 6 days. One patient discontinued PNS due to discomfort during stimulation [16]. Two studies investigated the placement of PNS after total knee arthroplasty (TKA) in 10 patients [24,25]. Implants were placed between 8 and 97 days from the original surgery. Patients reported significant reductions in pain levels during periods of rest, with an average decrease of 93% and 63% [13,14]. Similarly, pain reductions were noted during both active (average of 30% and 50%) and passive ranges of motion (average of 27% and 14%) [24,25]. PNS did not affect the maximum range of motion in these studies [24,25]. One study examined the use of a permanent PNS placement in 2 patients with chronic knee pain following revision TKAs [23]. Prior to implantation, patient 1 and 2 reported their pain as a 10 and an 8 on a scale of 0–10, respectively [23]. Four leads targeting the articular and cutaneous branches of the knee were placed. Patients improved in walking distance, standing tolerance, sitting tolerance, and sleep quality. Pain reduction ranged from 50% to 90% [23]. One patient discontinued hydromorphone because of her pain relief [23]. The major limitation of these studies was the lack of a control group.

One randomized control crossover case series discussed the use of PNS for pain management in 10 patients undergoing anterior cruciate ligament (ACL) reconstruction [15]. Leads targeting the femoral nerve were placed 2 days before surgery in all patients, removed prior to surgery, and then reconnected after surgery [26]. Preoperatively, a patient-controlled perineural catheter was inserted into the adductor canal as rescue analgesia. Ten patients were randomly separated into two groups of 5 patients, one receiving electrical stimulation and the other receiving a sham for 5 min. The groups were then switched for another 5 min interval [26]. Subjects initially experiencing stimulation experienced a reduction in surgical pain over 5 min, while the sham group reported an upward trend in pain. [26] This group continued to experience a downward trend during the next 5 min (crossover period). There was an average of 84% reduction in knee pain following 5 min of PNS in both groups [26]. Stimulation was administered for 30 min after the crossover period. After stimulation was turned off (average of 33 min following baseline), 50% of patients requested supplemental opioids and 70% initiated the continuous adductor canal block prior to discharge [26]. The major limitation of this study relates to the carryover effect of PNS, which subjects the patient to a continued variable duration and degree of analgesia following PNS discontinuation, possibly due to sustained modification of supraspinal pain processing. This would make the data of the 5-min sham period difficult to interpret. Nevertheless, in order to keep the study design this was included.

Lastly, one original study examined the efficacy of PNS in orthopedic trauma by assessing its impact on patients undergoing TKA as a proxy for military trauma [22]. They argue that orthopedic trauma induced during total knee replacement surgery best represents the traumatic combat-related orthopedic injuries experienced by military members because the TKA involves major injury to the joint and surrounding tissue, which can produce prolonged pain and extended opioid use [22]. The study evaluated the use of auricular nerve targeted PNS in 2 patients following total knee or hip arthroplasty, as a surrogate for battlefield trauma [22]. The device was placed on the posterior aspect of the right ear on two patients in under 3 min in the recovery room and was removed without complications on day 5 [22]. Patients had no pain while lying, sitting, or ambulating but reported pain as 2 to 5 out of a 0 to 10 pain scale while lowering onto the toilet [22]. Both patients experienced an increase in overall pain after the auricular nerve stimulator was removed [22]. The selection criteria for this study were not specified.

### 3.2. Shoulder

Six articles studied the use of PNS for shoulder pain (Table 2) [29,30,31,32,33,34]. Indications for PNS included subacromial impingement syndrome, chronic shoulder pain, operative pain, and osteoarthritis. The axillary nerve and suprascapular nerves were targeted with temporary and permanent devices in 23 patients. All studies reported pain outcomes based on subjective pain scales including the brief pain inventory short form, a numerical pain rating scale, and the visual analog scale [29,30,31,32,33]. Five studies reported the frequency, amplitude, and duration of the PNS treatment, which were in the ranges of 1.5–12 Hz, 0.2–20 mA, and 30–200 microseconds, respectively. 

In one study by Wilson et al., 10 patients with subacromial impingement syndrome failing six months of conservative therapy (defined as a trial of physical therapy and at least one subacromial steroid injection) were enrolled in the trial [29]. PNS was administered daily for six hours for a total of three weeks. Patients were followed for 12 weeks for pain monitoring. Statistically significant reductions in shoulder pain were seen at weeks 5, 8, and 16 compared to baseline levels of 46%, 50%, and 58%, respectively (*p* < 0.05) [29]. One patient was lost during follow up and two withdrew after the second month due to pain relief following subacromial steroid injections. There were no complications [29]. There was a 45% average improvement in disability at the end of treatment (EOT) (*p* < 0.01), a 48.6% increase in active shoulder ROM (*p* < 0.01), and an improved quality of life score in 80% of patients at EOT [29]. This study however lacked a control group, had short follow ups, lacked accounting for analgesic consumption, and could have a placebo effect.

Three studies reported pain outcomes in 11 patients with chronic shoulder pain due to a history of rotator cuff tear, arthritis, biceps tendinopathy, and/or adhesive capsulitis [30,31,33]. In a study by Ycaza and Vanquathem, one patient with chronic shoulder pain underwent implantation of permanent PNS leads after a temporary PNS trial [30]. Device settings were set based on trial settings and patient preference. At 1 year post operation, the patient continued to have almost 100% pain relief and successfully decreased his opioid intake from 15 mg 3 times per day to twice a day. The patient had complete relief from shoulder pain but continued to have pain from other comorbidities [30]. The study did not address the selection criteria or the limitation of having only one subject. A second study evaluated the effect of alternative cyclic sensory and motor stimulation of eight patients suffering from chronic shoulder pain [31]. Patients received a 4 h sensory stimulation followed by a 15 min rest period and then 1 h motor stimulation. The pain rating was based on the Numerical Pain Rating Scale (NPRS) with a baseline pain level of 8.14. At the mean follow up period (445 days) the average NPRS score was 2.7 (66% improvement) [31]. Sixty-two point five percent of patients who used opioids prior to PNS decreased their consumption after PNS treatment [31]. This study was retrospective and lacked a control group. The third study analyzing PNS and chronic shoulder pain due to rotator cuff pathology followed two patients unresponsive to conservative management [32]. The intervention targeted both the suprascapular and axillary nerves. Both patients experienced significant pain relief with an average of 90% relief reported during treatment and 85% pain relief 3 months after lead removal [32]. Both patients reported decreased opioid intake by 62%. No adverse events or complications were reported by the patients in any of the three studies discussed [32]. However, these patients initially refused surgical intervention, which could lead to a placebo effect or selection bias. 

One case report followed a patient who underwent PNS lead implantation prior to the conversion of total shoulder arthroplasty to a reverse total shoulder arthroplasty to control postoperative pain [33]. Preoperatively, a stimulating nerve block catheter was inserted into the shoulder and the PNS device was coiled and connected to the catheter. The stimulator settings were tuned to a frequency of 3 Hz, an amplitude of 0.5 mA, and a constant pulse duration of 0.3 ms. Following proper tuning of the device settings to achieve appropriate pain relief, the patient underwent a 6 h reverse total shoulder arthroplasty [33]. In the immediate postoperative period, the patient denied shoulder pain or discomfort. Unfortunately, later on postoperative day 1, the patient admitted increased pain in the shoulder, rating it an 8/10 on the numerical pain rating scale [33]. While her pain was adequately controlled with analgesics, it was found that the nerve block catheter had become dislodged and the PNS device was not able to effectively manage her pain any longer [33]. However, these patients initially refused surgical intervention, which could lead to a placebo effect or selection bias.

Lastly, one study used PNS therapy for the treatment of chronic shoulder pain following total shoulder replacement [22]. After an initial 60 day PNS trial, the patient reported 70% pain relief over 2 months and elected to proceed with permanent lead implantation. The device settings alternated between 20 Hz–8 Hz frequency and 6 mA–11 mA amplitude for 1–4 h each day to provide personalized pain relief. With permanent lead implantation, the patient again reported greater than 70% pain relief for 8 months. There were no reported complications [34]. This patient was selected because other treatment options for pain control were not feasible.

### 3.3. Foot

Two studies demonstrated effective pain control using PNS in the acute pain management following foot surgery (Table 3). In 2018, Ilfeld et al. published a proof-of-concept randomized control trial analyzing the use of PNS in patients undergoing hallux valgus osteotomy [35]. Leads targeting the sciatic nerve were placed in seven subjects one week prior to surgery in the treatment (n = 4) and sham group (n = 3). Sham stimulators were programmed and visually identical to the active stimulators. The stimulator was disconnected until after surgery when stimulation was delivered to the treatment group (n = 4) at a frequency of 100 Hz, an amplitude of 0.2–20 mA, and a pulse duration ranging from 15–200 µs [35]. After 5 min, the intervention was delivered to the sham group, and after 10 min, both groups received PNS. After 30 min, all leads were replaced with perineural catheters delivering opioid medication until removal on postoperative day (POD) 1 to 3. In both groups, PNS resulted in a downward trend of surgical pain, which continued to decrease to an average of 52% of baseline during the 30 min of stimulation and reduced opioid consumption [35]. Adverse events included cramping, lead dislodging, and lead fracture, none of which cause serious injury [35]. The major limitation of this study relates to the continued analgesic affect occurring after the washout period, which could possibly confound results.

A second proof-of-concept case series by Ilfeld et al. in 2022 studied the effect of percutaneous auricular nerve stimulation following ambulatory orthopedic and breast surgery in 7 patients [36]. PNS was placed in 4 patients following either bunionette and hammertoe correction, Haglund excision or hammertoe correction surgery. Although the results of all 7 patients were compiled together, the average rated daily pain at rest and while moving was a median of 0–1 on a scale of 0 (no pain) to 10 (most pain) [36]. Five patients avoided opioid use while 2 consumed 5 mg of oxycodone during POD 1 to 2. These cases demonstrate the feasibility and possible effectiveness of auricular percutaneous PNS in the treatment of ambulatory orthopedic surgery [36]. Patients in this study were offered PNS after surgery, which could have affected patient selection.

## 4. Discussion

The purpose of this study is to summarize the articles discussing the use of PNS in orthopedics and offer a comprehensive conclusion about its role. PNS is a relatively new application of neuromodulation that has been shown to be an effective treatment for pain. The current literature demonstrates that PNS has the potential to be an adequate pain reliever in a variety of orthopedic settings. The methodologies of the included studies vary, with case series being more robust than case reports. However, all studies reported pain relief following PNS, with one study reporting statistically significant results. Some studies also reported reduced opioid consumption because of PNS, which may be beneficial for select patient populations. However, conclusions regarding opioid consumption in the setting of PNS cannot be made. Adverse effects were minor and mostly consisted of lead fracture, cramping, and discomfort. No cases of infections were present. Theoretically, placement of leads close to blood vessels or nerves can damage them during insertion. Lead expulsion can also occur since muscle contraction has the tendency to repel foreign bodies outward.

A multicentered, randomized, double blind placebo-controlled study by Goree et al., published after data was abstracted for this study, assessed the impact of a 60 day PNS treatment of persistent postoperative pain after TKA (n = 41). Both the sciatic and femoral nerves were targeted. In this study, 60% of participants receiving PNS experienced pain reduction by 50% or more compared to 24% in the placebo group (*p* = 0.028). The PNS group also walked further in the 6 min walk test (47% vs. 9%, *p* = 0.048). This study suggests that PNS decreases persistent pain and improves functional outcomes, results similar to the studies mentioned in this systematic review [37]. In contrast, this study has more subjects and a more robust methodology. Similar methods can be used to expand the application of PNS to more than just postoperative surgical pain following TKA. Future studies should focus on outcomes of PNS in patients with severe osteoarthritis and contraindications to surgery or comparisons of pain relief efficacy between PNS or opioid medication.

PNS may be an effective pain relief modality for patients diagnosed with knee osteoarthritis and contraindications to surgery due to young age, comorbidities, or elevated BMI. This application of PNS addresses a quandary in knee arthroplasty for which there is currently no optimal solution. For these patients, PNS can provide a means for temporary pain relief by allowing patients the time to optimize their medical comorbidities and improve their candidacy for joint replacement surgery. If suitable, permanent lead placement can follow temporary PNS trials, offering a viable means of pain relief. These patients can prolong TKA or even avoid surgery. However, research in this area is limited. Theoretically, PNS may benefit these patients but practically it is impossible to tell the risks or benefits of this procedure. PNS may have a similar application in patients with shoulder osteoarthritis or persistent pain following total shoulder arthroplasty. Surgery in this population is often driven by pain, which can be managed by PNS.

Additionally, PNS may reduce the perception of pain in patients with severe knee deformities, but they do not correct the contractures, weakness, and instability of degenerative joint disease. By alleviating the sensation of pain, patients may feel more comfortable to perform high risk tasks that would otherwise be thwarted by the sensation of pain. On the contrary, patients treated with PNS may be more amenable to physical therapy and muscle strengthening, which can increase stability and improve their risk of falls. Ultimately, research analyzing the long-term effects of PNS in patients with knee or shoulder osteoarthritis in the context of rehabilitation is necessary before it becomes an integrated part of treatment. 

A few articles have examined the application of PNS in orthopedic trauma and ambulatory foot surgery. Among these, those addressing orthopedic trauma have employed patients undergoing TKA as a proxy due to the variability in trauma patient presentation. While TKA may bear some resemblance to the experiences of trauma patients, prospective pragmatic studies assessing the use of auricular PNS in patients arriving in the trauma bay may offer a more precise representation of this application. Auricular nerve stimulation can be applied quickly without an incision, which makes it ideal for trauma patients that may have multiple injuries. Although sample sizes are small and follow up is short, PNS has also shown benefits in ambulatory foot surgery.

While this systematic review is reflective of the current literature, the studies included are subject to selection bias and a placebo effect, which can lead to confounding. The absence of randomization and comparative methodologies with controls also hinders the formulation of conclusive findings. Therefore, long-term prospective reviews (with temporary and permanent leads) need to be performed in order to better understand adverse events, indications, outcomes, and complications of PNS in orthopedics. The result of this systematic review supports that PNS may be advantageous in the management of pain in orthopedics, but again, more robust studies are needed before it can be implemented.

This systematic review and the included articles have several limitations and shortcomings. Firstly, the studies consisted solely of case reports or case series, both of which are characterized by low statistical power, complicating the derivation of precise conclusions. Not all studies incorporated randomization or comparison groups, rendering the results less credible due to the incomparability of the groups. Numerous abstracts were discovered without complete manuscripts; these were omitted because they provided insufficient information. Pain and functional outcomes were measured using various metrics, muddling comparisons and hindering statistical analysis. The system’s technical constraints, such as lead fractures and dislodgement, could impact patient satisfaction and the effectiveness of the treatment. Additionally, the targeting of disparate nerves contributes to the study’s confounding variables. Finally, all studies published by Ilfeld received funding and support from the device manufacturer, introducing potential bias.

The existing literature on PNS has several gaps. Firstly, no study has thoroughly investigated the impact of PNS on intraoperative anesthesia, surgical recovery time, or rehabilitation. Additionally, the psychological effects of pain and their influence on patient outcomes have not been adequately addressed. Lastly, understanding the interaction between PNS and pain medications, including nonsteroidal anti-inflammatory drugs and opioids, is crucial. These investigations are important to understand the clinical application of PNS.

## 5. Conclusions

PNS has the potential to be an effective treatment modality to control pain in a variety of orthopedic settings and conditions. However, the literature only consists of case reports and case series which limits the power of this systematic review and increases the risk of bias. Further studies encompassing larger samples, randomized methods, and longer follow ups are necessary to establish its risks and benefits before its application can be ubiquitous.

## Figures and Tables

**Figure 1 jcm-13-03699-f001:**
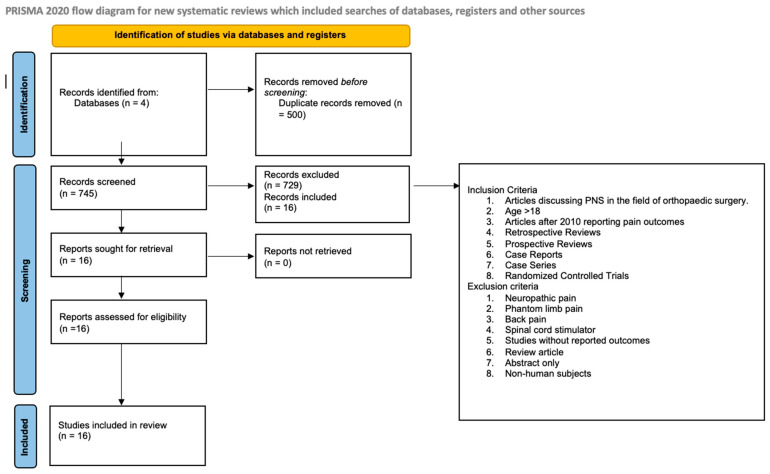
PRISMA guidelines and procedures used to extrapolate articles for this systematic review. Seven hundred and forty five (745) articles were screened and a total of 16 articles were included [18].

**Table 1 jcm-13-03699-t001:** (**a**) Summary of studies evaluating the use of PNS in patients with knee osteoarthritis. (**b**) Summary of studies evaluating the use of PNS in patients undergoing TKA.

(**a**)								
**Year**	**Author**	**Study Type**	**Patients**	**Indication**	**Nerves Targeted**	**Pain Prior to PNS**	**Pain Immediately after PNS**	**Pain 8 Weeks after PNS**
2022	Zhu CC, Gargya A, Haider N. [28]	Case series	3	Osteoarthritis of the knee	Femoral and saphenous	Patient 1: 7/10	Patient 1: 2/10	Patient 1: 4/10
					Femoral and saphenous	Patient 2: 7/10	Patient 2: 0/10	Patient 2: 2/10
					Saphenous	Patient 3 *: 6/10	Not assessed	Not assessed
2021	Hasoon J, Chitneni A, Urits I, Viswanath O, Kaye AD [27]	Case series	1	Osteoarthritis of the knee	Saphenous	Patient 4: 10/10	100% relief	Not assessed
(**b**)								
**Year**	**Author**	**Study Type**	**Patients**	**Indication**	**Nerves Targeted**		**Outcome**	
2019	Ilfeld BM, Ball ST, Gabriel RA et al. [16]	Case series: feasibility study	7	Postoperative knee pain following TKA	Femoral and sciatic		Average pain at week 1: <4/10 in 6 out of 7 patients	Time to preoperative walking level: 2 weeks
2017	Ilfeld BM, Grant SA, Gilmore CA, et al. [24]	Case series	5	Postoperative knee pain following TKA	Femoral: anterior knee pain Sciatic: posterior knee pain		63% average decrease in pain at rest 14% average pain reduction with passive ROM 50% average pain reduction with active ROM	
2017	Ilfeld BM, Gilmore CA, Grant SA, et al. [25]	Case series	5	Postoperative knee pain following TKA	Femoral and sciatic		93% average decrease in pain at rest 27% average pain reduction with passive ROM 30% average pain reduction with active ROM	
2010	McRoberts WP, Roche M [23]	Case series	2	Postoperative knee pain in patients with revision TKA’s	Articular and cutaneous branches of the knee		Patient 1: 50–70% pain reduction during the day. 100% pain reduction at night	Patient 2: complete pain relief and 80–90% pain relief at 2½ months
2019	Ilfeld BM, Said ET, Finneran JJ 4th, et al. [26]	Randomized control crossover case series	10	Postoperative knee pain following ACL reconstruction	Femoral		84% average reduction in knee pain following 5 min of stimulation	
2022	Ilfeld BM, Finneran JJ 4th, Said ET, Cidambi KR, Ball ST. [22]	Case series	2	TKA as a surrogate for trauma	Auricular		No pain with lying, sitting, or ambulating Mild pain while lowering onto the toilet. Reoccurrence of pain after implant removal.	

* Patient discontinued PNS after two weeks due to cramping and inadequate pain area coverage.

**Table 2 jcm-13-03699-t002:** Summary of studies evaluating the use of PNS in patients with shoulder pain.

Year	Author	Study Type	Patients	Indication	Nerves Targeted	Outcome
2014	Wilson RD, Harris MA, Gunzler DD, Bennett ME, Chae J. [29]	Case series	10	Subacromial impingement syndrome	Axillary nerve	36.6% pain relief. 45.5% improvement in DASH *. 52% reduction in pain interference. 48.6% improvement in active ROM.
2022	Ycaza R, Vanquathem N. [30]	Case report	1	Chronic shoulder pain, rotator cuff tear	Suprascapular nerve	Decreased pain medication intake and 100% pain relief at 1 year post-op.
2020	Mansfield JT, Desai MJ [31]	Case series	8	Chronic shoulder pain	Axillary nerve	Average pain reduction was 67%. All patients who used opioids prior to PNS decreased their original dosage.
2022	Chitneni A, Hasoon J, Urits I, Viswanath O, Berger A, Kaye AD [32]	Case series	2	Chronic shoulder pain	Suprascapular and axillary nerve	Average of 90% pain relief during treatment, 85% pain relief at 3 months after lead removal. There was a 62% decrease in opioid intake.
2022	Sondekoppam RV, Jindal A, Ip V, Tsui BCH [33]	Case report	1	Post-arthroplasty pain control	Axillary nerve	Postoperatively controlled pain but the PNS catheter became dislodged.
2020	Mansfield JT, Desai MJ [34]	Case report	1	Post-arthroplasty pain control	Axillary Nerve	Pain remained low at 8 months post-op.

* = Shoulder-related disability index.

**Table 3 jcm-13-03699-t003:** Outcomes of patients undergoing foot surgery treated with PNS.

Year	Author	Study Type	Patients	Indication	Nerves Targeted	Pain Immediately After PNS	Pain 8 Weeks After PNS
2022	Ilfeld BM, Gabriel RA, Said ET, et al. [35]	Randomized control trial	7	Postoperative pain following Hallux valgus osteotomy	Sciatic	Patient 1: 2/10 Patient 2: 0/10 Not assessed	Patient 1: 4/10 Patient 2: 2/10 Not assessed
2021	Ilfeld BM, Finneran JJ, Dalstrom D, Wallace AM, Abdullah B, Said ET [36]	Case series	4	Postoperative pain control following ambulatory foot surgery	Auricular	100% relief	0/10

## Data Availability

The original data presented in the study are openly available in references [16,17,18,19,20,21,22,23,24,25,26,27,28,29,30,31,32,33,34,35,36].

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
