# Peer review of "The Relevance of Implanted Percutaneous Electrical Nerve Stimulation in Orthopedics Surgery: A Systematic Review"

_jcm, 2024, doi:10.3390/jcm13133699_

Round 1
Reviewer 1 Report
Comments and Suggestions for Authors
I would like to thank the authors for giving me the opportunity to review this paper. Systematic reviews can be tough to carry out and write. The authors' efforts do not go unnoticed.
I think the current SR has some significant methodological flaws. First, the authors report they followed the PRISMA guidelines, but they do not cite these standards and do not provide a PRISMA checklist. There are several instances where the guidelines are clearly not followed. Examples include, but are not limited to: not including "A Systematic Review" in the title, being more explicit in their inclusion/exclusion criteria [PICOTS criteria are standard], outlining the synthesis methods and risk of bias assessments which were planned (if any), etc. The authors also do not report whether the SR was registered prospectively/retrospectively (e.g., PROSPERO), so I am assuming it is not. The authors do not report whether a meta-analysis was planned, or was unable to be carried out and why.
These methodological issues make the results and conclusions difficult to ascertain and follow given the uncertainty of the original intent of the review, and the manner in which it was carried out.
Comments on the Quality of English Language
There are several instances of the authors not using the PNS abbreviation once it is established. There are a total of 10 instances where this needs to be corrected.
Author Response
Reviewer 1
I would like to thank the authors for giving me the opportunity to review this paper. Systematic reviews can be tough to carry out and write. The authors' efforts do not go unnoticed.
- I think the current SR has some significant methodological flaws. First, the authors report they followed the PRISMA guidelines, but they do not cite these standards and do not provide a PRISMA checklist.
- Checklist included
- There are several instances where the guidelines are clearly not followed. Examples include, but are not limited to: not including "A Systematic Review" in the title, being more explicit in their inclusion/exclusion criteria [PICOTS criteria are standard], outlining the synthesis methods and risk of bias assessments which were planned (if any), etc.
- Systematic review was added to the title. PICOT criteria was added along with synthesis methods. The study quality assessment using the questionnaire provided by murad et al. for case series and reports were added as well. Risk of bias and quality assessment was also included in the manuscript.
- The authors also do not report whether the SR was registered prospectively/retrospectively (e.g., PROSPERO), so I am assuming it is not. The authors do not report whether a meta-analysis was planned, or was unable to be carried out and why.
- Study was not registered on PROSPERO because authors were not aware of this prior to conducting the study. Studies with data already abstracted cannot be registered on PROSPERO. Meta Analysis could not be carried out because outcomes varied between studies and reporting was not standardized. This was added to the manuscript.
- These methodological issues make the results and conclusions difficult to ascertain and follow given the uncertainty of the original intent of the review, and the manner in which it was carried out.
- The original intent to review is discussed in the introduction and discussion. The original intent is to summarize the literature regarding the use of PNS in the field of orthopedic surgery and provide a comprehensive conclusion about its role. Based on the results, the evidence is sparse and cannot provide robust and substantial conclusions about its role in orthopedics. Although systematic reviews are conducted to help form definitive conclusions, this is not possible because the literature is not supportive of that. However, after synthesis of data the results of PNS in the field of orthopedic surgery look promising and has the potential to help manage pain in certain patient population.
- There are several instances of the authors not using the PNS abbreviation once it is established. There are a total of 10 instances where this needs to be corrected.
- Abbreviations were replaced for full terms after the abbreviation was introduced.
Thank you for your feedback. It provided helpful insight and made us improve our study design and purpose to make it better.
Reviewer 2 Report
Comments and Suggestions for Authors
The purpose of your systematic review is a broad challenge and tries to summarize the literature involving peripheral nerve stimulation in orthopedic surgery. You must ensure that your systematic review covers all necessary aspects of PNS use in orthopedics, from scientific mechanisms to practical applications and ethical considerations. Please, ensure the review is comprehensive and considers all relevant scientific perspectives:
1. The review must consider the impact of PNS on surgical recovery times compared to traditional pain management methods. This point is important because understanding recovery times can help determine the practical benefits of PNS. If this data is not available in the current literature, it should be noted as a gap in research.
2. You could cite the specific mechanisms of PNS that are thought to affect the sensory and motor fibers in the context of neuropathic pain. This is critical for understanding how PNS could potentially help patients with neuropathic symptoms, which are common in orthopedic conditions. The review should discuss these mechanisms.
3. Are there any comparative studies on PNS versus opioid management for postoperative pain in orthopedics included in the review? Comparative effectiveness is key to validating PNS as a viable alternative to opioids. If such studies are lacking, this should be identified as a research need.
4. Integrating PNS with physical therapy and rehabilitation protocols for orthopedic patients could optimize recovery. The absence of this discussion would be a significant gap, suggesting a need for interdisciplinary research.
5. Technical limitations can affect the applicability of study results. Cite what are the technological limitations of the PNS devices used in the studies reviewed, and how might they affect outcomes. This aspect should be discussed to guide future device improvements.
6. Cite how PNS influences the need for anesthesia during orthopedic procedures. Are there studies comparing intraoperative PNS use? This question explores the potential of PNS to reduce anesthesia use, which is relevant for reducing complications and recovery time.
7. Were the methodologies of the included studies sufficiently robust to support the conclusions drawn in the review? This critique helps ensure to the readers that the review's conclusions are based on strong evidence. If methodologies are weak, this should be openly discussed.
8. The review could consider the psychological effects of chronic pain management with PNS. Understanding the psychological outcomes is crucial for holistic treatment. If not discussed, it highlights a need for more comprehensive patient care studies.
9. Ethical considerations are fundamental in clinical trials and treatments. What ethical considerations are discussed regarding the implementation of PNS, especially related to patient consent and potential risks? If the review lacks this discussion, it misses a critical aspect of clinical implementation.
10. Is there information on the interaction of PNS with medications commonly used in orthopedic patients, particularly NSAIDs and opioids? Drug-device interactions are essential for patient safety and efficacy of treatment. The review should address this, or acknowledge the gap.
11. Identifying biases in study selection is crucial for the credibility of the review. If not assessed, the review should mention this as a limitation. The authors must address the bias in the selection of studies included in the review, such as publication bias or language bias. There are many others. These limitations can affect how much the readers will trust the results of your review. It’s important to address these limitations issues to provide stronger evidence about the effectiveness of PNS in orthopedic surgery and stimulate the research to these gaps:
a) Many studies included in your review had a small number of participants. Most are small sample sizes. This can make it hard to say if the results will be the same for more people.
b) The studies often had short follow-up times with patients. This makes it difficult to know if the benefits or problems from the treatment last a long time.
c) Some studies did not randomly assign patients to treatment groups. This can make the results less reliable because the groups may not be comparable at the start.
d) Many studies didn’t compare PNS to other treatments. Without comparisons, it's hard to tell if PNS is better, worse, or the same as other options.
e) The review included many case reports and series, which are types of studies that don’t provide very strong evidence. These studies can tell us what happened to one person or a small group, but they don't show a clear cause and effect.
f) The review did not assess the risk of bias within the studies. This means there might be factors in the studies that could make the results seem better or worse than they really are.
g) It appears that the review excluded abstracts and studies without full reports. Some important findings might have been missed because these types of studies were not considered.
Author Response
The purpose of your systematic review is a broad challenge and tries to summarize the literature involving peripheral nerve stimulation in orthopedic surgery. You must ensure that your systematic review covers all necessary aspects of PNS use in orthopedics, from scientific mechanisms to practical applications and ethical considerations. Please, ensure the review is comprehensive and considers all relevant scientific perspectives:
- The review must consider the impact of PNS on surgical recovery times compared to traditional pain management methods. This point is important because understanding recovery times can help determine the practical benefits of PNS. If this data is not available in the current literature, it should be noted as a gap in research.
- The following was added “No study considered its impact on surgical recovery time which illustrates a gap in the research.”
- You could cite the specific mechanisms of PNS that are thought to affect the sensory and motor fibers in the context of neuropathic pain. This is critical for understanding how PNS could potentially help patients with neuropathic symptoms, which are common in orthopedic conditions. The review should discuss these mechanisms.
- The mechanism of action is described in the introduction. Neuropathy is complex and is not commonly managed solely by orthopedic surgeons. The authors decided to exclude neuropathy in this study. Reintroducing neuropathy would alter the study in its entirety. The authors do understand that the review is broad, but literature suggesting its use in orthopedic surgery is sparse.
- Are there any comparative studies on PNS versus opioid management for postoperative pain in orthopedics included in the review? Comparative effectiveness is key to validating PNS as a viable alternative to opioids. If such studies are lacking, this should be identified as a research need.
- Unfortunately, there are no comparative studies between PNS and opioids. The studies included in this review report some data regarding decreased opioid use but none are comparative.
- The following was added” The absence of randomization and comparative methodologies with controls also hinders the formulation of conclusive findings.”
- Integrating PNS with physical therapy and rehabilitation protocols for orthopedic patients could optimize recovery. The absence of this discussion would be a significant gap, suggesting a need for interdisciplinary research.
- The authors agree and this was modified “No study considered its impact on surgical recovery time or incorporation with rehabilitation which illustrates a gap in the research.”
- Technical limitations can affect the applicability of study results. Cite what are the technological limitations of the PNS devices used in the studies reviewed, and how might they affect outcomes. This aspect should be discussed to guide future device improvements.
- The following was added “There are several gaps in the literature affecting the clinical application of PNS.
- No study considers the impact of PNS on intraoperative anesthesia, surgical recovery time or rehabilitation.”
- Cite how PNS influences the need for anesthesia during orthopedic procedures. Are there studies comparing intraoperative PNS use? This question explores the potential of PNS to reduce anesthesia use, which is relevant for reducing complications and recovery time.
- No study compares the effect pf PNS and intraoperative anesthesia and its affect on PNS. The following was added “There are several gaps in the literature affecting the clinical application of PNS. No study considers the impact of PNS on intraoperative anesthesia, surgical recovery time or rehabilitation.”
- Were the methodologies of the included studies sufficiently robust to support the conclusions drawn in the review? This critique helps ensure to the readers that the review's conclusions are based on strong evidence. If methodologies are weak, this should be openly discussed.
- The following was clarified. The methodologies of the included studies vary, with case series being more robust than case reports. However, all studies reported pain relief following PNS, with one study reporting statistically significant results.
- Strengths and Weakness of the methodologies are discussed in the discussion. The methods of the included studies are robust to support the conclusion, which states that “PNS has the potential to be an effective treatment modality to control pain in a variety of orthopedic settings and conditions”. Due to the lack of robust comparative studies with larger sample size definitive conclusions cannot be made. But the compilation and analysis of all 16 studies suggests that the potential of PNS in orthopedics is greater than any individual study.
- The review could consider the psychological effects of chronic pain management with PNS. Understanding the psychological outcomes is crucial for holistic treatment. If not discussed, it highlights a need for more comprehensive patient care studies.
- The following was modified and added “There are several gaps in the literature affecting the clinical application of PNS. No study considers the impact of PNS on intraoperative anesthesia, surgical recovery time or rehabilitation. Moreover, the psychological impact of pain and its effect on patient outcomes is absent in the literature as well.
- Ethical considerations are fundamental in clinical trials and treatments. What ethical considerations are discussed regarding the implementation of PNS, especially related to patient consent and potential risks? If the review lacks this discussion, it misses a critical aspect of clinical implementation.
- The following was added to the methods “All case series included were IRB approved. Patient consent was obtained in all studies”
- Is there information on the interaction of PNS with medications commonly used in orthopedic patients, particularly NSAIDs and opioids? Drug-device interactions are essential for patient safety and efficacy of treatment. The review should address this, or acknowledge the gap.
- The following was modified and added“ The existing literature on PNS has several gaps. Firstly, no study has thoroughly investigated the impact of PNS on intraoperative anesthesia, surgical recovery time, or rehabilitation. Additionally, the psychological effects of pain and their influence on patient outcomes have not been adequately addressed. Lastly, understanding the interaction between PNS and pain medications, including nonsteroidal anti-inflammatory drugs and opioids, is crucial. These investigations are important understand the clinical application of PNS.
- Comparative studies with pain medication are not present in the literature just yet.
- Identifying biases in study selection is crucial for the credibility of the review. If not assessed, the review should mention this as a limitation. The authors must address the bias in the selection of studies included in the review, such as publication bias or language bias. There are many others. These limitations can affect how much the readers will trust the results of your review. It’s important to address these limitations issues to provide stronger evidence about the effectiveness of PNS in orthopedic surgery and stimulate the research to these gaps:
- a) Many studies included in your review had a small number of participants. Most are small sample sizes. This can make it hard to say if the results will be the same for more people.
-Agreed. This is included as a limitation and is discussed in the manuscript.
- b) The studies often had short follow-up times with patients. This makes it difficult to know if the benefits or problems from the treatment last a long time.
-Agreed. This is also included as a limitation. These studies analyzed temporary PNS implants which lasts approximately 60 days. Therefore, follow up greater than 60 days is unlikely to be present in the literature unless the study analyzes permanent PNS.
- c) Some studies did not randomly assign patients to treatment groups. This can make the results less reliable because the groups may not be comparable at the start.
-This was addressed in the limitations portion of the discussion
- d) Many studies didn’t compare PNS to other treatments. Without comparisons, it's hard to tell if PNS is better, worse, or the same as other options.
- This was addressed in the limitations portion of the discussion
- e) The review included many case reports and series, which are types of studies that don’t provide very strong evidence. These studies can tell us what happened to one person or a small group, but they don't show a clear cause and effect.
- This was addressed in the limitations portion of the discussion
- f) The review did not assess the risk of bias within the studies. This means there might be factors in the studies that could make the results seem better or worse than they really are.
-Biases were included at the end of each study or studies. The quality assessment tool specific to case series and case reports by murad et al was used and included in the supplemental data. The authors agree that bias is present. Unfortunately, the literature only consists of case series and case reports at this time.
- g) It appears that the review excluded abstracts and studies without full reports. Some important findings might have been missed because these types of studies were not considered.
-Authors agree. This was included in the limitations. Abstracted were excluded because of the lack of data available for these abstracts.
Thank you for your feedback. It provided helpful insight and made us improve our study design and purpose to make it better.
Reviewer 3 Report
Comments and Suggestions for Authors
The topic is undoubtedly interesting and very relevant, especially with a large number of patients with joint pain. Reducing pain using direct nerve stimulation reduces the need for the use of NSAIDs and opioid drugs. By reducing the side effects of drugs and achieving a pronounced analgesic effect, the patient’s activity and quality of life improve several times.
Based on the results of the work, it is clear that the authors mean electrical stimulation. In this case, electrical nerve stimulation has 3 types. Transcutaneous electrical nerve stimulation, percutaneous electrical nerve stimulation and implanted percutaneous electrical nerve stimulation. It would be more correct if the author changed the title to “Relevance of implanted percutaneous electrical nerve stimulation in orthopedics surgery”. I understand that the term "PNS peripheral nerve stimulation" is widely used. however, it is difficult to know what type of stimulation is being used. Many studies use transcutaneous electrical nerve stimulation to treat joint pain. Percutaneous electrical nerve stimulation is also used using an inserted needle, which is removed after the procedure.
In lines 42-44. It cannot be said that electrical nerve stimulation is based on the gate control theory proposed by Melzack and Wall in 1965, since this method has been around for a long time before the creation of this theory. It would be more correct to say that the pathogenetic basis of electrical nerve stimulation was substantiated using the theory of gate control of pain.
In lines 48-50. Experimental and clinical studies on electrical nerve stimulation were conducted until 2022. Especially the works of Sluka E. It would be right to start with these works.
In lines 51-54, the authors stated that the generator controls the frequency, amplitude and duration of the electrical stimulation pulse to provide individualized treatment. In fact, the current characteristics are determined by the physiotherapist, not the device. It is important to note that electrical nerve stimulation uses two types of electrical impulses. High-frequency low-amplitude and low-frequency high-amplitude current depending on the required therapeutic effect. Implanted transcutaneous electrical nerve stimulation usually uses pulses of high frequency and low amplitude. explain this in your work.
Add more information about another analgesic effects of percutaneous electrical nerve stimulation beyond the gate theory of pain.
The purpose of this systematic review must be clarified. To summarize the literature involving implanted percutaneous peripheral electrical nerve stimulation in orthopedic surgery
In table 1: You indicated “Zhu AC” the author of three scientific papers. Although this author's manuscript is a review. Please indicate each author of the works mentioned [Zhu AC, Recommendations for anesthetic management for intraoperative neuromodulation cases. Pain manag. 2022;12(4):557-567. doi:10.2217/pmt-2020-0109].
Incorrect reference numbers in Table 2. Mansfield et al [18]. Chitneni et al [20] and in table 2 Ilfeld et al18 [18].
References are not formatted according to journal rules.
It is difficult to assess the side effects of implanted percutaneous electrical nerve stimulation based on case data. However, moving the needle close to the nerve and large vessels can damage them during limb movement. In addition, the implanted electrode cannot remain in the same place where it was inserted for a long time. This fact is due to the fact that muscle tissue has the ability to repel a foreign body outward. We often see this with electroacupuncture or percutaneous electrical nerve stimulation. Without a doubt, these facts may limit the use of this treatment in clinical practice. Not bad if you have studied these points in the discussion.
Author Response
Reviewer 3
The topic is undoubtedly interesting and very relevant, especially with a large number of patients with joint pain. Reducing pain using direct nerve stimulation reduces the need for the use of NSAIDs and opioid drugs. By reducing the side effects of drugs and achieving a pronounced analgesic effect, the patient’s activity and quality of life improve several times.
- Based on the results of the work, it is clear that the authors mean electrical stimulation. In this case, electrical nerve stimulation has 3 types. Transcutaneous electrical nerve stimulation, percutaneous electrical nerve stimulation and implanted percutaneous electrical nerve stimulation. It would be more correct if the author changed the title to “Relevance of implanted percutaneous electrical nerve stimulation in orthopedics surgery”. I understand that the term "PNS peripheral nerve stimulation" is widely used. however, it is difficult to know what type of stimulation is being used. Many studies use transcutaneous electrical nerve stimulation to treat joint pain. Percutaneous electrical nerve stimulation is also used using an inserted needle, which is removed after the procedure.
- The title was changed.
- In lines 42-44. It cannot be said that electrical nerve stimulation is based on the gate control theory proposed by Melzack and Wall in 1965, since this method has been around for a long time before the creation of this theory. It would be more correct to say that the pathogenetic basis of electrical nerve stimulation was substantiated using the theory of gate control of pain.
- The authors agree and this was added. “The pathogenetic basis of electrical nerve stimulation was substantiated using the theory of gate control of pain a theory proposed by Melzak and Wall in 1965 which explains that the sensory input by large Aβ fibers obstructs the transmission of input from the small pain fibers”.
- In lines 48-50. Experimental and clinical studies on electrical nerve stimulation were conducted until 2022. Especially the works of Sluka E. It would be right to start with these works.
- The following was added “The clinical and experimental use of electrical stimulaiton for pain releif is a field of medicine that continues to grow.”
- The following was added “The clinical and experimental use of electrical stimulaiton for pain releif is a field of medicine that continues to grow.”
- In lines 51-54, the authors stated that the generator controls the frequency, amplitude and duration of the electrical stimulation pulse to provide individualized treatment. In fact, the current characteristics are determined by the physiotherapist, not the device. It is important to note that electrical nerve stimulation uses two types of electrical impulses. High-frequency low-amplitude and low-frequency high-amplitude current depending on the required therapeutic effect. Implanted transcutaneous electrical nerve stimulation usually uses pulses of high frequency and low amplitude. explain this in your work.
- The following was modified and added “ These are programmed by the person providing the device. The application of PNS typically involves using a continuous biphasic waveform with a pulse duration ranging from 250 to 500 ms, and it can be delivered at either low or high frequency.
- Moreover, specific settings used are described in the results
- Add more information about another analgesic effects of percutaneous electrical nerve stimulation beyond the gate theory of pain.
- The following was modified and added “The pathogenetic basis of electrical nerve stimulation was substantiated using the theory of gate control of pain a theory proposed by Melzak and Wall in 1965 which explains that the sensory input by large Aβ fibers obstructs the transmission of input from the small pain fibers. This theory suggests that stimulation of large nonnociceptive afferent fibers (pressure, touch, vibration) will “close” the dorsal horn interneurons and inhibit pain perception, whereas small nociceptive afferent fibers opens the dorsal horn interneurons.[2–8] In other words, the low activation threshold for the large Aβ fibers causes early activation and resultant inhibition of nociceptive Aδ and C pain fibers by exciting the associated dorsal horn interneuron involved in processing and transmitting pain.[5,9] PNS works by providing a nonpainful stimulation to a peripheral nerve resulting in an increased electrical threshold, decreased excitability, and slowing conduction velocity that inhibits pain. [2–8] Electrical stimulation of these fibers can downregulate neurotransmitters (substance P and calcitonin gene related-peptide [CGRP]), endorphins and local inflammatory mediators to associated with pain and a nerves sensory distribution.[5,9] It is the release of these neurotransmitters that augments the pain response leading to vasodilation, plasma extravasation, macrophage chemotaxis, and/or mast cell degranulation.[6,10]”
- The purpose of this systematic review must be clarified. To summarize the literature involving implanted percutaneous peripheral electrical nerve stimulation in orthopedic surgery
- The following was modified The purpose of this systematic review is to summarize the literature involving PNS in orthopedic surgery and provide a comprehensive conclusion about its role”
- In table 1: You indicated “Zhu AC” the author of three scientific papers. Although this author's manuscript is a review. Please indicate each author of the works mentioned [Zhu AC, Recommendations for anesthetic management for intraoperative neuromodulation cases. Pain manag. 2022;12(4):557-567. doi:10.2217/pmt-2020-0109].
- All authors were added according to the references
- Incorrect reference numbers in Table 2. Mansfield et al [18]. Chitneni et al [20] and in table 2 Ilfeld et al18 [18].
- References were modified.
- References are not formatted according to journal rules.
- References were modified
- It is difficult to assess the side effects of implanted percutaneous electrical nerve stimulation based on case data. However, moving the needle close to the nerve and large vessels can damage them during limb movement. In addition, the implanted electrode cannot remain in the same place where it was inserted for a long time. This fact is due to the fact that muscle tissue has the ability to repel a foreign body outward. We often see this with electroacupuncture or percutaneous electrical nerve stimulation. Without a doubt, these facts may limit the use of this treatment in clinical practice. Not bad if you have studied these points in the discussion.
- This was added “Theoretically, placement of leads close to blood vessels or nerves can damage them during insertion. Moreover, it is difficult for leads to stay in place since muscle has the tendency to repel foreign bodies outward”
Thank you for your feedback. It provided helpful insight and made us improve our study design and purpose to make it better.
Round 2
Reviewer 1 Report
Comments and Suggestions for Authors
I would like to thank the authors for their revisions to this SR.
Unfortunately, I think there are still methodological errors, mostly related to PRISMA guidelines not being followed throughout the design and carrying out of the review, or the reporting of this compliance being unclear. Reporting of the methodology remains unclear. Examples include, clearly stating the PICO question instead of stating PICO was applied, describing how was data extracted and whether this done in duplicate (what form was used to collect the data), a listing of excluded studies after full text review and why they were excluded, etc.
My previous comment on the original intent of the review being unclear is related to the lack of prospective registration of the protocol for the review, not the introduction and discussion of the paper being unclear. It is impossible to know whether the research question, aims, etc. are consistent with those originally set forth at the beginning of the review.
Reviewer 2 Report
Comments and Suggestions for Authors
The revisions made have significantly strengthened the overall quality of the manuscript. The revised manuscript now stands as a more robust and coherent contribution to the field. The attention to detail and the thoughtful incorporation of feedback demonstrate your commitment to producing a high-quality scientific work. I would like to express my appreciation for the diligence and care you and your co-authors have taken in addressing the comments and making necessary corrections. Don't worry about the literature gaps, this is one of the article's greatest contributions to the community, I hope it serves as an encouragement to other researchers who have the same passion for the topic. And consequently they will cite this publication.
Reviewer 3 Report
Comments and Suggestions for Authors
Congratulations to the authors for improving the manuscript. Of course, we really need this kind of work to develop this area. Electrical nerve stimulation in all its aspects is evolving daily and is increasingly used in the treatment of many diseases. If the beginning of the technique was associated with the treatment of pain, then recently it has been used in the treatment of many orthopedic diseases. In its modern form, the manuscript is of a high level and is worthy of publication in such a rated journal. The topic has not been thoroughly studied and I hope that after this work there will be other studies studying this treatment method on a large sample size. I am pleased to support this work and recommend it for publication.